# Genetic Diversity Analysis of Non-Heading Chinese Cabbage of Resistance to Clubroot Disease Based on SNP Molecular Markers

Liming Miao [1], Dandan Xi [1], Xiaofeng Li [1], Changwei Zhang [2], Lu Gao [1], Ying Li [2], Yuying Zhu [1] and Hongfang Zhu [1,*]

1 Shanghai Key Laboratory of Horticultural Protected Horticultural Technology, Horticultural Research Institute, Zhuanghang Comprehensive Experiment Station, Shanghai Academy of Agricultural Sciences, Shanghai 201403, China
2 State Key Laboratory of Crop Genetics and Germplasm Enhancement, College of Horticulture, Nanjing Agricultural University, Nanjing 210095, China
* Correspondence: zhuhongfang@saas.sh.cn

**Abstract:** Clubroot disease is a kind of soil-borne disease that seriously infects *Brassica* species. In this study, we collected 121 varieties of non-heading Chinese cabbages. In order to better understand the genetic variation and to screen suitable clubroot disease-resistant parental material, we re-sequenced them to examine the population genetic structure, population genetic diversity, population differentiation index, and selective sweep based on SNPs. The mapping rate with the reference genome was high, and data quality analysis revealed that the sequencing quality was good. The annotated data indicated that intronic and intergenic areas held the majority of SNPs and indels. Four subgroups of 121 non-heading Chinese cabbages were identified using principal component analysis, phylogenetic tree, and genetic structure analysis. An examination of genetic diversity revealed that while selfing may happen in subgroups C and D, heterozygosity may exist in subgroups A and B. In subgroup B, self-fertilization is not possible. There was a moderate degree of genetic differentiation between subgroups B and C (Fst = 0.0744347). For genes in certain sweep regions, we also ran GO enrichment and KEGG enrichment analysis. Two disease resistance-related genes, *BraA01g042910.3.5C* and *BraA06g019360.3.5C*, were examined. These findings will serve as a theoretical foundation for developing novel, clubroot disease-resistant types of non-heading Chinese cabbages.

**Keywords:** non-heading Chinese cabbage; clubroot disease; SNPs; genetic diversity; selective sweep

## 1. Introduction

Non-heading Chinese cabbage (*Brassica campestris* ssp. *chinensis* L.) is an annual or biennial herbs native to China with a long history of cultivation. It belongs to the genus *Brassica* and the family Cruciferae. The middle and lower portions of the Yangtze River and its southern parts adore it for its tender quality, distinct flavor, abundant nutrition, and other qualities. At present, the current farmed area in China is around 1.3333 million hm$^2$, which is significant to the annual production and supply of vegetables. Clubroot disease is a kind of soil-borne disease caused by *Plasmodiophora brassicae*. Currently, clubroot disease is the most significant soil-borne disease of *Brassicaceae* crops in the world and results in significant economic losses every year [1–3]. Low antigen and low resistance levels have been the main limiting factors for breeding brassica crops for clubroot disease resistance. At present, Chinese cabbage is one of the most studied brassica crops in the field of clubroot disease. Multiple resistance sites have been reported in the genome of Chinese cabbage, but only two resistance genes have been cloned, namely *CRa* and *Crr1a*, which are *R*-genes with NBS LRR [4,5]. Soil affected by *P. brassicae* is challenging to eradicate because it can persist in the soil as dormant spores for decades [6]. Clubroot disease can be lessened through chemical control; however, when chemicals are misused, the environment can be easily contaminated and the physical and chemical composition of the soil altered. Currently, the

most affordable, efficient, and environmentally friendly method of controlling clubroot disease is through the breeding of new varieties [7].

The foundation of ecological and species variety is genetic diversity, which is typically defined as the sum of genetic variation between various populations within a species or between various individuals within a population [8]. The foundation of high-quality novel variety breeding is the germplasm resources. An efficient way to find superior breeding materials and boost breeding effectiveness is to analyze the genetic diversity of a large number of germplasm resources. Molecular markers are effective tools to detect the genetic diversity of germplasm resources. Restriction fragment length polymorphisms (RFLP) is the earliest molecular marker technique [9], and Random amplified polymorphic DNA (RAPD) [10], Amplified fragment length polymorphism (AFLP), also known as Selective Fragment Selection amplification (SRFA) [11], these are considered to be the first generation of molecular markers. Subsequently, SSR and ISSR combined with PCR technology are considered second-generation markers and are widely used [12]. Single nucleotide polymorphisms (SNP) are known as third-generation DNA molecular markers, which are currently being employed extensively to research population genetic evolution [13–15]. For instance, Song et al. utilized SNP markers to accurately identify the genotypes and analyze the genetic diversity of 114 Chinese jujube germplasm [16]. In addition, SNP markers have also been applied to study genetic diversity in *Brassicaceae* crops [17–21]. We collected 121 varieties, but the genetic diversity among these varieties was unknown. In this study, the incidence of clubroot disease in these materials was studied, the genetic relationship and population structure of different resistance cultivars were analyzed based on SNP markers. These results provide a basis for breeding and germplasm innovation of high-quality disease-resistant non-heading Chinese cabbage.

## 2. Material and Methods

### 2.1. Plant Materials

The progeny of anti-clubroot disease Chinese cabbage and non-heading Chinese cabbage were backcrossed with non-heading Chinese cabbage. The materials with agronomic traits that favored non-heading Chinese cabbage were selected and separated by self-cross for five generations, and sixty-one materials were obtained for sequencing. Sixteen sequencing materials were derived from non-heading Chinese cabbage and loose-leaf Chinese cabbage from Japan and South Korea after six generations of self-cross separation. Twenty-three of them were identified by manual identification method as anti-sensitive materials after self-cross separation, and twenty-one of them were high-generation homozygous materials owned by our research group. Non-heading Chinese cabbages were sown in the hole tray containing *P. brassicae* on 5 September 2020. The incidences of all plants were identified on 3 November 2020. Each variety was transplanted according to the number of seedlings, with the largest number of 63 and the smallest number of 2. The incidence of each individual plant was observed, and then the incidence of each variety was counted. All plants were recolonized in greenhouses in the Fengpu campus of the Shanghai Academy of Agricultural Sciences. The young leaves were collected and quick-frozen in liquid nitrogen and then stored in a −80 °C refrigerator on 5 January 2021.

### 2.2. Resequencing and Data Quality Control

The DNA of 121 Chinese cabbage was tested by Nanodrop and agarose gel electrophoresis and then a next-generation sequencing library was constructed. The whole genome resequencing was performed by Tianmen Hanlin Gene Technology Co., Ltd. (Wuhan, China) with Illumina HiSeq 3000 after the library quality was qualified. FASTP was used to filter raw data according to the following criteria: (1) If one end is low-quality reads, the reads paired with it are removed; (2) the reads with quality value $Q \leq 15$ accounting for 40% of the total bases are removed; (3) the reads containing more than 5 N are removed; (4) the reads with length less than 15 bp are removed; (5) Splice sequences at both ends of reads were removed [22]. FastQC (http://www.bioinformatics.babraham.

ac.uk/projects/fastqc, accessed on 14 June 2022) was applied to analyze the base quality distribution. Base content distribution, GC content distribution and sequencing base quality were also analyzed in this study.

### 2.3. Sequence Alignment and SNP/Indel Detection

In this study, *Brassica rapa* Chiifu V3.5 (http://brassicadb.cn/#/Download/, accessed on 14 June 2022) was selected as the reference genome, and BWA MEM was utilized to align clean data to the reference genome [23]. Then, GATK MarkDuplicates (version 4.1.2.0) was used to filter repeated reads. GATK HaplotypeCaller (version 4.1.2.0) (in GVCF mode) was employed to detect and filter SNP/Indel (Table S1). ANNOVAR software was used to annotate SNP/indels and perform statistics [24].

### 2.4. Group Evolution Analysis

PLINK (http://pngu.mgh.harvard.edu/~purcell/plink/, accessed on 14 June 2022) was chosen for population structure analysis. Firstly, the Ped file of PLINK was created, and then ADmixture software was applied to construct the population genetic structure and population lineage information. For the study population, the number of subgroups (K value) was set as 1–10 in advance for clustering, and the clustering results were cross-validated, and the optimal number of clusters was determined according to the valley value of the cross-validation error rate. Distances between populations were calculated by VCF2Dis (https://github.com/BGI-shenzhen/VCF2Dis, accessed on 14 June 2022). Using Treebest (http://treesoft.sourceforge.net/treebest.shtml, accessed on 14 June 2022) software to calculate distance matrix. On this basis, a phylogenetic tree was constructed by the neighbor-joining method. Bootstrap values were calculated 1000 times. Smartpca package in EIGENSOFT (https://www.hsph.harvard.edu/alkes-price/software/, accessed on 14 June 2022) was used to conduct PCA analysis. The population's genetic diversity was analyzed by using the command as populations in the Stacks program package. Fst analysis indicates the degree of population differentiation. The larger the value, the higher the degree of population differentiation and the higher the degree of selection. VCFTools (—FST-window-size 200,000—FST-window-step 20,000) was selected to calculate the Fst value in this study. The selective sweep regions were analyzed by vcftools-v0.1.16 based on Fst with 200 kb as the size of the sliding window.

### 2.5. Linkage Disequilibrium Analysis

Linkage disequilibrium (LD), also known as allelic association, refers to the non-random association between two alleles on the same chromosome. Generally, LD intensity is related to the distance between two SNPs. The smaller the distance, the smaller the chance of recombination, and the stronger the LD. Conversely, the greater the distance, the greater the chance of recombination, and the weaker the LD. In this study, PopLDdecay (V3.41) was employed for LD analysis, the parameter was set to MaxDist 10000.

## 3. Results

### 3.1. Data Statistics and Evaluation

A total of 121 cabbages were sequenced, and the largest number of raw reads is from the plant numbered P21-1-249, with 34,566,744 raw reads and 34,538,110 clean reads after filtering. The least number of raw reads is from the plant numbered P21-1-93, with 12,680,398 raw reads and 12,672,024 clean reads. The mean number of raw reads obtained from all samples was 16,794,360 and the mean number of clean reads was 16,784,963 (Table 1). The analysis results of base quality distribution, base content distribution, GC content distribution and sequencing base quality showed that the sequencing quality was high and could be used for subsequent analysis. The filtered reads were compared to the reference genome of *Brassica rapa*, and the average alignment rate of all samples was 98.31%. The plant numbered P21-1-86 had the highest alignment rate, with 99.12%. The plant numbered P21-1-17 was the lowest, with an alignment rate of 91.31% (Table 1). The

average sequencing depth of all samples was 6.75X, and the highest was the P21-1-249, which was 13.85X. The lowest was P21-1-10, which was 4.92X. The mean coverage of 1X sequencing depth was 83.27%, the mean coverage of 5X sequencing depth was 37.15%, and the mean coverage of 10X sequencing depth was 7.74% (Table 1). The above results indicate that the sequencing quality of this study is high, the alignment rate is high, and the data is reliable, which can be further analyzed.

**Table 1.** Statistics and quality assessment of sequence data.

|  |  | Maximum Value | Minimum Value | Average Value |
|---|---|---|---|---|
| Reads statistic | Raw reads | 34,566,744 | 12,680,398 | 16,794,360 |
|  | Raw data bases | 5,185,011,600 | 1,902,059,700 | 2,519,154,000 |
|  | Clean reads | 34,538,110 | 12,672,024 | 16,784,963 |
|  | Clean data bases | 5,113,683,916 | 1,871,020,512 | 2,484,015,380 |
|  | Clean data Q20 (%) | 97.95 | 94.79 | 96.17 |
|  | Clean data Q30 (%) | 94.03 | 85.13 | 89.00 |
|  | Clean data GC (%) | 39.43 | 37.34 | 38.19 |
| Mapped reads statistic | Total Reads | 35,124,117 | 12,871,646 | 17,038,713 |
|  | Map Reads | 34,762,929 | 12,649,523 | 16,750,465 |
|  | Map Rate | 99.12% | 91.31% | 98.31% |
|  | UniqMap Rate | 95.96% | 87.69% | 94.54% |
| Sequencing depth | Coverage (%) | 90.53 | 77.02 | 83.27 |
|  | Coverage 5X (%) | 71.81 | 21.60 | 37.15 |
|  | Coverage 10X (%) | 34.65 | 3.60 | 7.74 |
|  | Depth | 13.85 | 4.92 | 6.75 |
|  | Coverage Base | 319,702,365 | 271,985,352 | 294,071,230 |

Q20: base ratio with quality value greater than 20; Q30: ratio of bases with quality value greater than 30; GC: GC content; Coverage: ratio of 1X coverage; Coverage 5X: ratio of 5X coverage; Covrage10X: ratio of 10X coverage; Depth: average sequencing Depth; Coverage Base: The number of bases of 1X covering.

### 3.2. SNP and Indel Detection and Analysis

The detected SNPs and indels were filtered and counted, and the highest number of loci consistent with the reference genome was P21-1-88 (132,447 SNPs and 23,499 indels), and the least number was P21-1-5 (40,558 SNPs and 7875 indels). The plant with the most homozygous loci was P21-1-80 (18,230 SNPs and 1841 indels), and the least was p21-1-86 (6610 SNPs and 778 indels). The plant with the largest number of heterozygous loci was P21-1-249 (51,452 SNPs and 5288 indels), which had the least number of deletion loci and total SNPs. The number of heterozygous loci was the lowest in P21-1-5 (10,189 SNPs and 1005 indels), which had the largest number of deletion loci and total SNPs (Table 2). The annotation result showed that 86,312 SNPs/indels were located in intergenic regions, 34,420 SNPs/indels were annotated in intronic regions, and there were also a large number of SNPs/indels in exons and upstream and downstream regions of genes, with fewer SNPs/indels in the 5′-UTR and 3′-UTR regions (Figure 1).

### 3.3. Population Genetic Structure and Phylogenetic Tree Analysis

Population genetic structure refers to the non-random distribution of genetic variation in a species or population. The subgroup to which the individual belongs can be determined through an association study of the genotype and phenotype of different individuals in the same subgroup that are closely related to each other. In this study, the number of subgroups was prespecified as 10 for clustering (K = 10), and the clustering results showed that when K = 4, the value of CV error is the smallest, which is 0.53272, so the optimal number of clusters is determined as 4. Subgroup A contained 53 varieties, which was the most abundant subgroup. There were nine members in subgroup B, which was the smallest subgroup. Subgroups C and D contained 23 and 36 varieties, respectively (Figure 2). Principal component analysis (PCA) showed that members of the four subgroups were clustered together, indicating that all samples can be divided into four subgroups (Figure S1A). The phylogenetic tree analysis showed that not all members were clustered

in one branch, for example, 19 members of group A were not clustered together with the others. P21-1-9, P21-1-14 and P21-1-15 have long genetic distances from the others in group D. On the whole, most members of the same subgroup in the four subgroups could be clustered together (Figure S1B). The incidence rates of clubroot disease of different varieties in different subgroups were calculated. In subgroup B and subgroup C, one of the varieties had a high incidence rate, while the other varieties had no significant difference. In subgroup A and subgroup D, the incidence rates of different varieties were quite different (Figure 3; Table S2). We demonstrated the phenotypes of clubroot disease of resistant cultivar P21-1-40 and susceptible cultivar P21-1-41. The susceptible cultivar had enlarged roots and weak plants (Figure 4).

**Table 2.** Statistics of SNPs and Indels.

|  |  | Sample | Maximum Value | Sample | Minimum Value | Average Value |
|---|---|---|---|---|---|---|
| SNP | Ref | P21-1-88 | 132,447 | P21-1-5 | 40,558 | 99,270 |
|  | Alt (Homo) | P21-1-80 | 18,230 | P21-1-86 | 6610 | 13,847 |
|  | Het | P21-1-249 | 51,452 | P21-1-5 | 10,189 | 28,723 |
|  | Miss | P21-1-5 | 118,656 | P21-1-249 | 3013 | 11,801 |
|  | Total | P21-1-5 | 177,601 | P21-1-249 | 136,896 | 153,641 |
| INDEL | Ref | P21-1-88 | 23,499 | P21-1-5 | 7875 | 18,460 |
|  | Alt (Homo) | P21-1-80 | 1841 | P21-1-86 | 778 | 1410 |
|  | Het | P21-1-249 | 5288 | P21-1-5 | 1005 | 2692 |
|  | Miss | P21-1-5 | 19,125 | P21-1-249 | 350 | 1657 |
|  | Total | P21-1-5 | 28,865 | P21-1-249 | 20,525 | 24,219 |

Ref: Number of loci consistent with the reference genome; Homo: number of homozygous sites; Het: number of heterozygous sites; Miss: number of missing sites; Total: indicates the number of all loci.

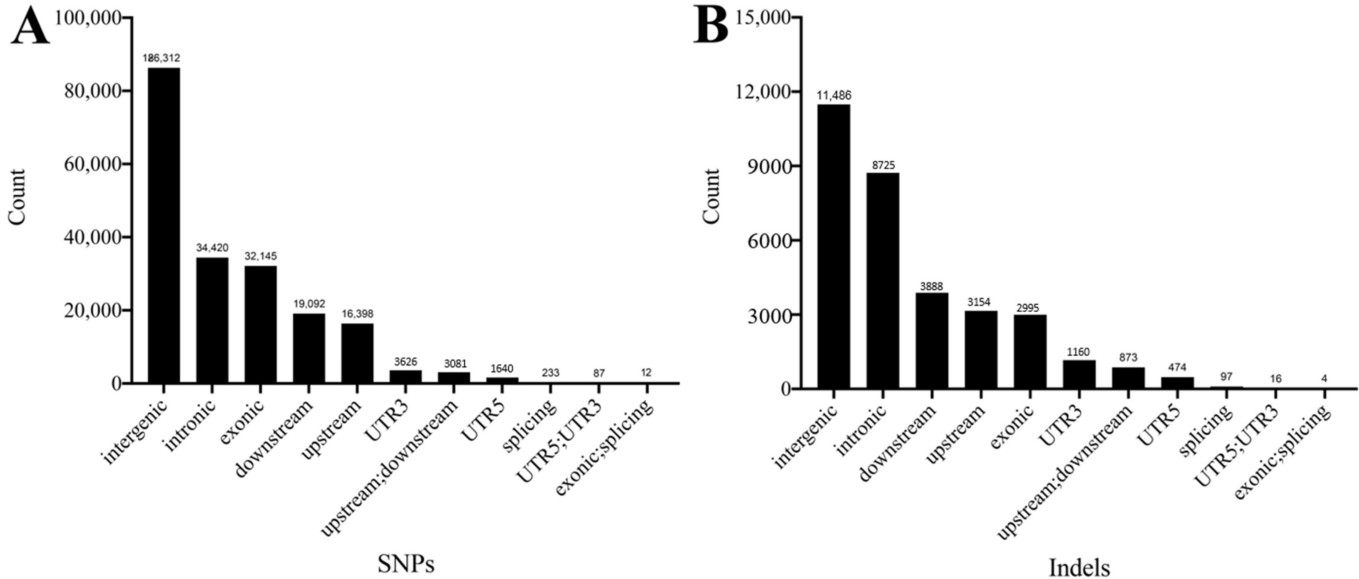

**Figure 1.** Statistics of SNPs (**A**)/Indels (**B**) annotations. Abscissa: annotated region of SNPs/Indels; Vertical coordinate: number of SNPs/Indels in different regions.

*3.4. Analysis of Population Genetic Diversity*

The results of the genetic diversity analysis of the four subgroups showed that the observed heterozygosity of subgroup A and subgroup B was greater than the expected heterozygosity, indicating that heterozygosity may exist in the subgroups. The observed heterozygosity of subgroup C and subgroup D was less than the expected heterozygosity, indicating that selfing might exist (Table 3). Nucleotide diversity ($\pi$) measures the genetic diversity within a population, and in this study, the highest genetic diversity was found in subgroup D ($\pi = 0.31612$) and the lowest was in subgroup B ($\pi = 0.23934$). The inbreeding number (Fis) is a measure of the degree of deviation from Hardy Weinberg within a

subgroup. The result showed that subgroup D has the highest degree of genetic diversity (Fis = 0.15415). The Fis value of subgroup B was negative (Fis = −0.0198), indicating that heterozygotes were predominant and homozygotes were absent in the subgroup (Table 3).

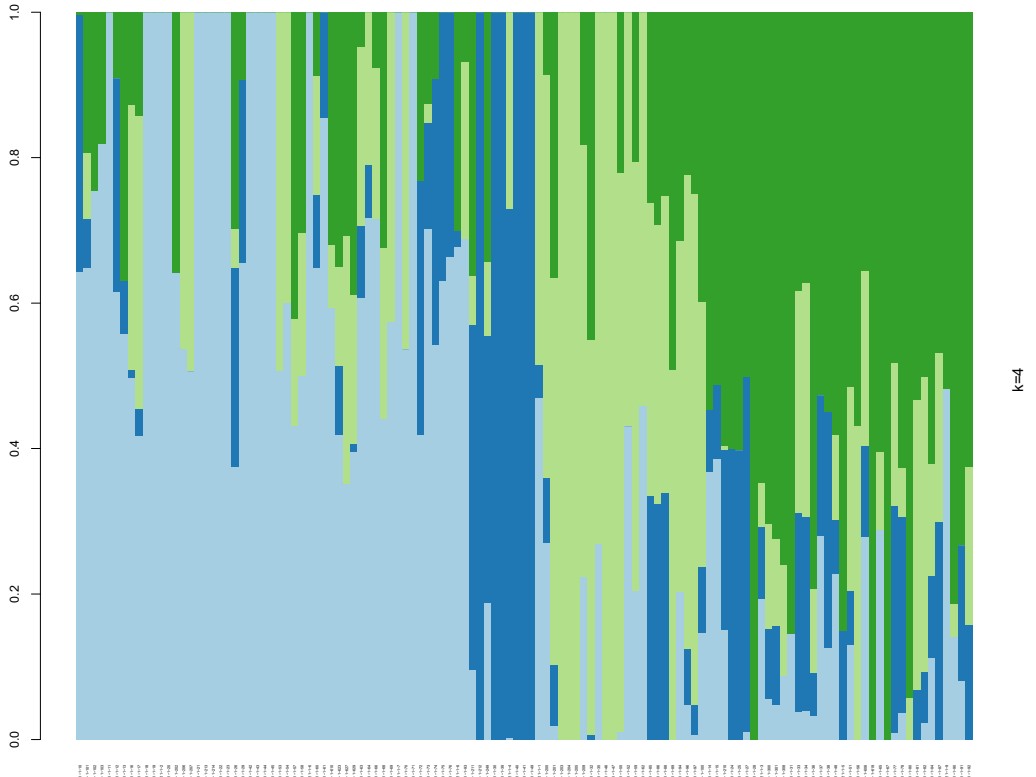

**Figure 2.** Population structure of 121 Non-heading Chinese cabbage.

### 3.5. Linkage Disequilibrium Analysis

Generally, D' and $r^2$ are used for testing. When $r^2 = 1$, it indicates that there is no recombination between the two sites. When $r^2 = 0$, it means that there is no LD or linkage equilibrium between the two sites. The higher the $r^2$ from 0 to 1, the higher the LD, and the higher the linkage. Subgroup B in this study had the highest degree of linkage; the $r^2$ values of subgroup A and subgroup D were similar, and the degree of linkage was relatively lower. The degree of linkage of subgroup C was slightly higher than that of subgroups A and D (Figure 5).

### 3.6. Selective Deletion Analysis

Fst represents the degree of differentiation between groups. The higher the value, the greater the differentiation and increased selection pressure. The value range is 0–1, with a maximum value of 1, indicating complete differentiation between the two groups and a minimum value of 0 indicating no differentiation between the groups. When Fst = 0~0.05, the differentiation is small and cannot be considered. When Fst = 0.05–0.15, there was a moderate degree of genetic differentiation among groups. When Fst = 0.15–0.25, the genetic differentiation among populations was great. When Fst > 0.25, there is great genetic differentiation among groups. In this study, there was a moderate degree of genetic differentiation between subgroups B and C (Fst = 0.0744347), and the other subgroups were all less than 0.05, indicating very little differentiation (Table 4).

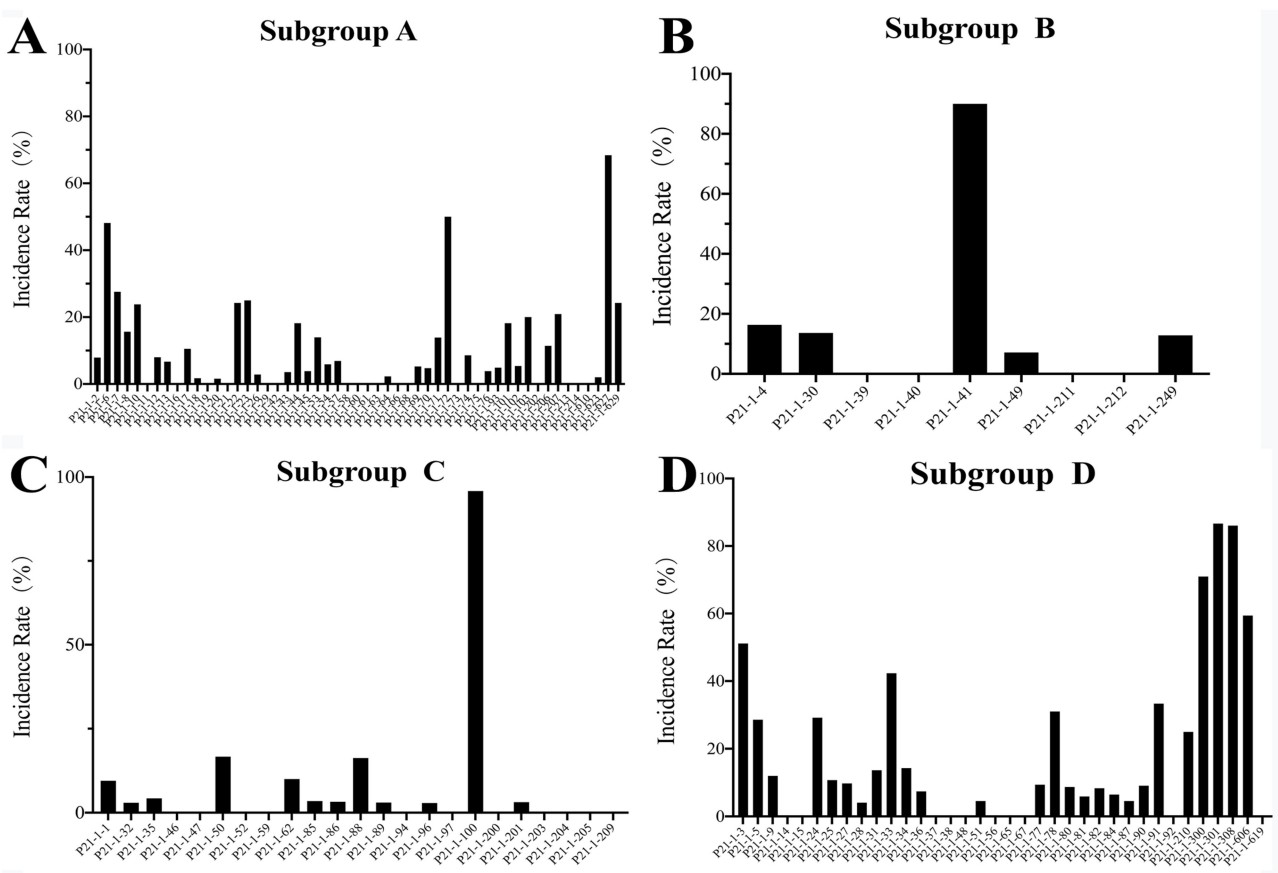

**Figure 3.** Statistics of incidence rates of different varieties in four subgroups. (**A**): The incidence rates of clubroot disease of different varieties in subgroup A; (**B**): The incidence rates of clubroot disease of different varieties in subgroup B; (**C**): The incidence rates of clubroot disease of different varieties in subgroup C; (**D**): The incidence rates of clubroot disease of different varieties in subgroup D.

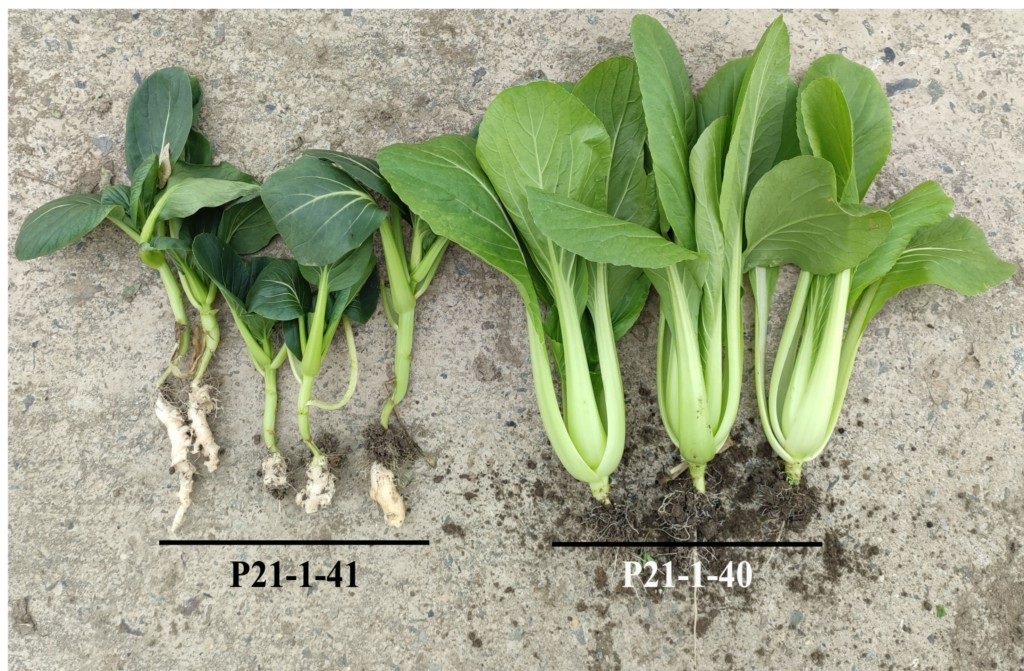

**Figure 4.** Phenotypes of clubroot disease in resistant variety P21-1-40 and susceptible variety P21-1-41.

**Table 3.** Analysis of population genetic diversity.

| Subgroup | Num_Indv | Obs_Het | Obs_Hom | Exp_Het | Exp_Hom | Pi | Fis |
|---|---|---|---|---|---|---|---|
| A | 50.77558 | 0.28931 | 0.71069 | 0.28849 | 0.71151 | 0.29138 | 0.07317 |
| B | 8.79188 | 0.2588 | 0.7412 | 0.22567 | 0.77433 | 0.23934 | −0.01984 |
| C | 22.03601 | 0.26015 | 0.73985 | 0.28823 | 0.71177 | 0.29497 | 0.12222 |
| D | 33.86832 | 0.26989 | 0.73011 | 0.31143 | 0.68857 | 0.31612 | 0.15415 |

Num Indv: average number of individuals per locus in the subgroup; Obs Het: observed heterozygosity of all loci in the subgroup; Obs Hom: average observed homozygosity of all loci in the subgroup; Exp Het: the average expected heterozygosity of all loci; Exp Hom: the average expected homozygosity of all loci in the subgroup; Pi: nucleotide diversity ($\pi$ value) of all loci in the subgroup; Fis: consanguinity coefficient of individuals within a subgroup.

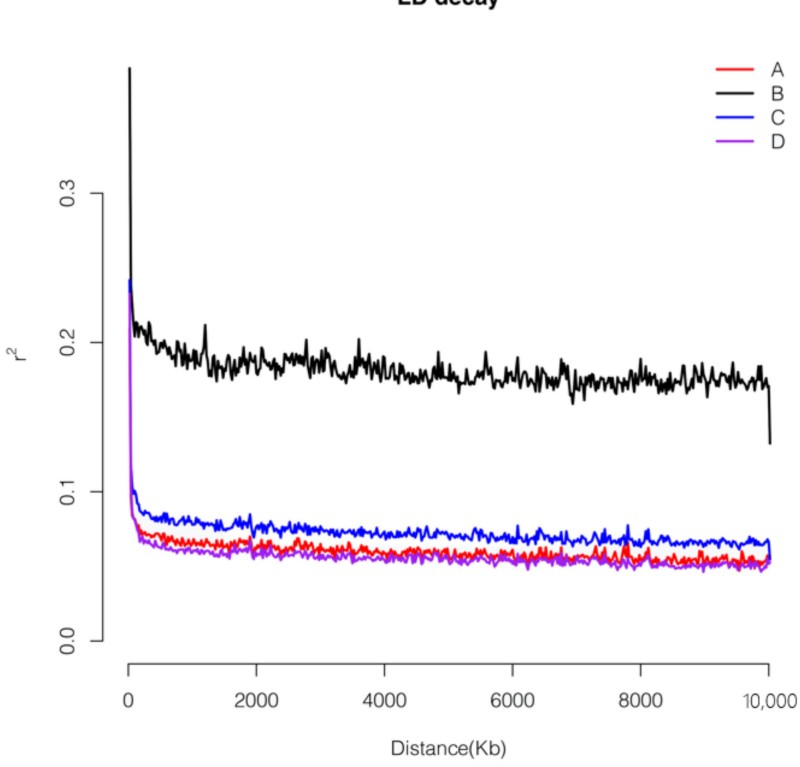

**Figure 5.** Linkage disequilibrium analysis. Abscissa: the clustering among SNPs; Vertical coordinate: $r^2$, a higher $r^2$ curve indicates that the varieties has a stronger LD.

**Table 4.** Population genetic differentiation index.

| | A | B | C | D |
|---|---|---|---|---|
| A | | 0.0455735 | 0.040152 | 0.0276534 |
| B | | | 0.0744347 | 0.0447235 |
| C | | | | 0.0321242 |

Fst = 0~0.05: differentiation is small and cannot be considered; Fst = 0.05–0.15: there was a moderate degree of genetic differentiation among populations. Fst = 0.15–0.25: greater genetic differentiation among populations; Fst > 0.25: very large genetic differentiation among populations.

In this study, selective clearance analysis was performed based on Fst. There was a moderate degree of genetic differentiation between subgroups B and C. There were 1255 annotated genes in the "selective sweep region" in the top 1% of Fst values between subgroup B and subgroup C. GO enrichment analysis was performed for these genes. In 'molecular function', genes are mostly enriched in 'catalytic activity', 'transporter activity' and 'binding'. When looking at 'biological processes', the top three organisms were 'metabolic process', 'cellular process' and 'single-organism process'; Among the 'cell components',

the top three were 'cell', 'cell part' and 'organelle' (Figure 6A). KEGG enrichment was performed and was divided into five categories for enrichment. These are 'Cellular Processes', 'Environmental Information Processing', 'Genetic Information Processing', 'Metabolism', and 'Organismal Systems'. Among the above five pathways, the most enriched pathways in each category were 'Transport and catabolism', 'Signal Transduction', 'Translation', 'Metabolism' and 'Environmental' Adaptation', respectively (Figure 6B). Among all the enriched genes, two genes were related to clubroot disease resistance, *BraA01g042910.3.5C* and *BraA06g019360.3.5C*. The KEGG annotation of BraA01g042910.3.5C is disease resistance protein RPM1, while BraA06g019360.3.5C is enhanced disease susceptibility protein EDS1 (Table S3).

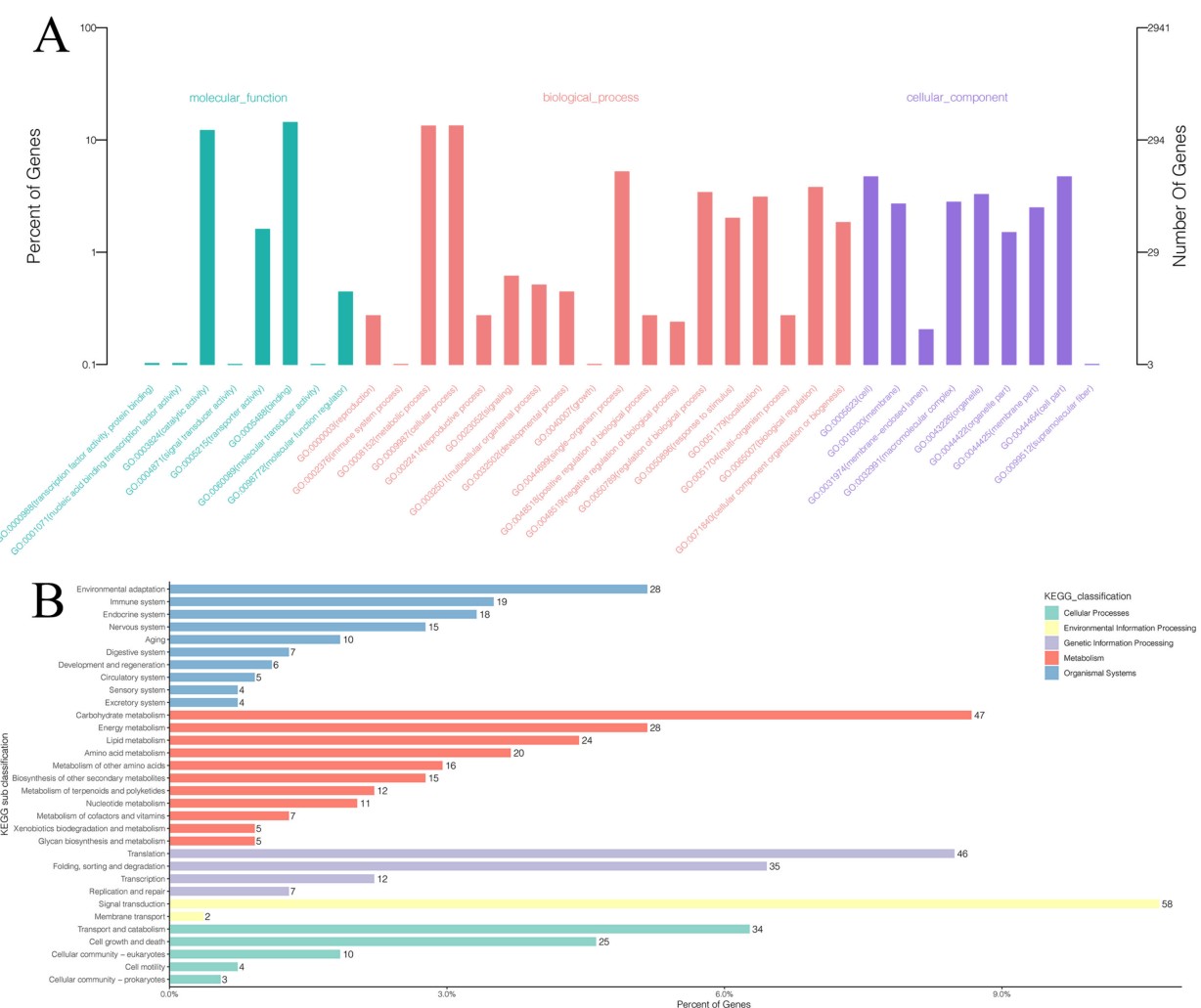

**Figure 6.** GO and KEGG enrichment of genes in the selective sweep regions between subgroup B and C. (**A**): GO enrichment; (**B**): KEGG enrichment.

## 4. Discussion

Both a species' genetic diversity and its population's genetic structure affect an organism's capacity for evolution as well as its resilience to harsh circumstances. The ability of a species to adapt to environmental changes is inversely correlated with the level of genetic diversity within that species, and genetic diversity loss frequently results in decreased adaptability, reproductive success, and disease resistance. As the most prevalent and stable kind of genetic variation in the majority of genomes, SNP markers are more reliable genetic markers [25,26]. In this work, the genetic diversity of 121 non-heading Chinese cabbages resistant to clubroot disease was investigated using SNP markers. There were 197,046 SNPs

found in total, with P21 having the most and P88 having the fewest. Intergenic areas were where most SNPs and indels were annotated.

One-hundred and twenty-one non-heading Chinese cabbages were separated into four categories in this study. Subgroup D has the highest genetic diversity ($\pi$ = 0.31612). There were 36 members in this category, and there were wide variations in the incidence rates of different varieties. This implies that one factor contributing to the varying incidence rates of various cultivars may be the genetic diversity within subgroups. Except for P21-1-41, subgroup B had the fewest members and the least genetic diversity ($\pi$ = 0.23934), and the incidence rates of the other variations were mostly unaltered. Additionally, subgroup B has a negative inbreeding rate. These might result from the small number of cultivars grouped together in this subgroup or the presence of self-incompatibility [27,28]. The genetic differentiation between groupings B and C in this study is moderate (Fst = 0.0744347), whereas the genetic differentiation between the remaining categories is minimal (Fst < 0.05). The findings of this study suggest that choosing between subgroups B and C when choosing parental materials could lead to increased heterosis. Genetic differentiation across groups is greatly influenced by gene flow [29], which is mostly accomplished by pollen and seed transfer [30,31]. The moderate degree of divergence between groupings B and C suggests that these two subsets frequently exchange genes.

During the process of domestication and selection, the genetic diversity of a species is reduced due to the directional selection of the mutation and the linked region of a certain trait, which is called 'Selective Sweep' [32–35]. It is simple to establish a strong "selective sweep effect" by choosing genes for crucial plant agronomic features [36]. In this study, groupings B and C differ genetically to a moderate extent. Based on the Fst values between the two subgroups, "selected sweep regions" were found. The genes in the "selected sweep regions" may contain candidate genes for clubroot resistance. In *Brassica rapa*, two CR genes were well-studied, *CRa* (A03) [4,37,38] and *Crr1a* (A08) [5]. In addition, many CR loci for genes whose functions have not been defined were also identified, such as *Crr3* [39], *Rcr9* [40], *Crd* [41], *CRs* [42] and others. However, the function of these genes needs to be further verified. In this study, two candidate genes related to disease resistance were found in the KEGG analysis, *BraA01g042910.3.5C* and *BraA06g019360.3.5C*. The KEGG annotation of BraA01g042910.3.5C is disease resistance protein RPM1, while BraA06g019360.3.5C is enhanced disease susceptibility protein EDS1. *RPM1* was one of the first genes for plant resistance to be discovered by spontaneous variation in *Arabidopsis thaliana* [43]. Nie et al. found that *TaRPM1* plays an important role in the mechanism of innate immunity to infection by the powdery mildew pathogen in wheat [44]. Enhanced Disease Susceptibility 1 (EDS1) family proteins, EDS1, Phytoalexin-Deficient 4 (PAD4) and Senescence-Associated Gene 101 (SAG101) are crucial regulators of effector-triggered immune (ETI) response and basal immunity [45,46]. However, the role of *RPM1* and *EDS1* in root disease resistance is unknown. GO enrichment revealed that "catalytic activity" and "binding" enriched more genes, while KEGG enrichment revealed that "signal transduction" and "carbohydrate metabolism" were the most enriched gene categories. Plants infected with *P. brassicae* will wither and even die [47]. Therefore, a large number of genes were enriched in carbohydrate metabolic pathways in KEGG analysis as expected. These results provide information for us to screen the genes of resistance to clubroot disease.

## 5. Conclusions

Population genetic structure, PCA, and phylogenetic tree analysis were used in this work to separate 121 non-heading Chinese cabbages into four subgroups. The greatest level of genetic diversity is found in subgroup D. Subgroups B and C differed genetically from each other to a moderate extent. Two genes of resistance to clubroot disease, *BraA01g042910.3.5C* and *BraA06g019360.3.5C*, were screened. These findings offered a theoretical basis for developing non-heading Chinese cabbage cultivars resistant to clubroot disease, and they also offered clues for the investigation of the molecular mechanisms behind clubroot disease resistance.

**Supplementary Materials:** The following supporting information can be downloaded at: https://www.mdpi.com/article/10.3390/agronomy13010245/s1, Figure S1: Principal component analysis (A) and evolutionary tree analysis (B). Table S1: The analysis software used in this study. Table S2: Statistics of incidence rates of different varieties. Table S3: KEGG annotation of genes in the selective sweep regions between subgroups B and C.

**Author Contributions:** H.Z. proposed and designed the research. D.X. and L.G. performed the investigation of the incidence rates of root swelling and sampling. Y.Z. and X.L. supervised this study, C.Z. and Y.L. reviewed this manuscript. L.M. performed the statistical analyses and wrote the manuscript. All authors have read and agreed to the published version of the manuscript.

**Funding:** This work was supported by Key science and technology project of Shanghai Science and Technology Commission (21N11900400). The funding agency had no role in the design, analysis, and interpretation of the data or writing of the manuscript, they provided financial support for our study.

**Institutional Review Board Statement:** Not applicable.

**Informed Consent Statement:** Not applicable.

**Data Availability Statement:** Plant materials are available under request to the respective owner institutions. The sequencing data presented in this study are openly available in NCBI under accession PRJNA907412.

**Conflicts of Interest:** The authors declare no conflict of interest.

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
