# Peer review of "Genetic Diversity Analysis of Non-Heading Chinese Cabbage of Resistance to Clubroot Disease Based on SNP Molecular Markers"

_agronomy, doi:10.3390/agronomy13010245_

Round 1

Reviewer 1 Report

Line 45: Please explain the following text: "ecological and species variety". Why reference 7 is important in this context?

Line 50: I'm not familiar with the expression: 3rd generation markers. Which markers are considered as first and second generation?

Lines 55-56: Previous study, where you selected plants used in this study, was mentioned but not cited.  From the information available in the article it is not clear how plants were obtained, how screening for the resistance was performed, was there any variability in resistance to clubroot disease, etc. Poorly written methods is the reason for introducing serious doubt in this study.

In addition, introduction is not well written. There is no information about currently known disease resistant genes.

Results of disease resistance screening for all 121 samples should be provided or citation to the article is these results were already published.

Line 71: What is BGI-T7? 

Line 83: Which version of GATK was used? Did you use GVCF mode?

Line 102: "LD is formed by mutation or recombination, which appears when a new mutation is produced near a SNP." I can not agree with this explanation. 

Line 114: Most of the sequencing datasets in SRA have from 6 to 10 M reads. Here you mentioned that minimum 12 M reads were obtained per sample. How the numbers you reported were obtained?

Additionally:

Did you observe any correlation between disease resistance and 4 groups obtained with clustering methods? I think information about plant phenotypes should be included in the results interpretation.

Reviewer 2 Report

The paper of Miao et al. presented the genetic diversity analysis of non-heading cabbage resistance to clubroot using SNP markers. The paper was well written and is qualified for publication after some minor revisions. I am attaching the file wherein you can view my comments and suggestion.

Reviewer 3 Report

Clubroot disease is the most dangerous soil-borne disease of genus Brassica in the world. The authors of the present paper performed the genetic diversity analysis of Chinese cabbage of resistance to clubroot disease based on SNP markers. They found two candidate genes related to disease resistance that may offer the theoretical foundation for breeding of disease resistant cultivars of Chinese cabbages.

Various statistical methods were used to interpret the results obtained. Color graphs also illustrate well the work done.

There are a few comments to the manuscript:

1)    Abstract. line 12. The authors wrote that they “screened 121 varieties of non-heading Chinese cabbages resistant to clubroot disease”. But according to the Figure 3 not all varieties were resistant. It is desirable that the authors give more information about the studied varieties.

2)    Abstract. lines 21-22. The authors concluded that “The genetic differences between subgroups B and C were quite minor.” But the genetic differences between another subgroups were even less according to the Table 4.

3)    Line 277. The authors wrote “there were wide variations in the incidence rates of different species” may be they mean “…of different varieties”.

Round 2

Reviewer 1 Report

The article has been greatly improved and many thanks to the authors for the answers.

I have three more comments:

line 94: sequencing machine - SRA database indicates that the sequences were obtained with Illumina HiSeq 3000. Was this an error during the submission?

line 156: Authors stated "The detected SNPs and indels were filtered...". A description of filtering step should be added to the methods.

Please include a description how selective sweep regions were identified in the methods section . How large the sliding windows were?
